# Overcoming the vanishing gradient problem in plain recurrent networks

## Abstract

Plain recurrent networks greatly suffer from the vanishing gradient problem while Gated Neural Networks (GNNs) such as Long-short Term Memory (LSTM) and Gated Recurrent Unit (GRU) deliver promising results in many sequence learning tasks through sophisticated network designs. This paper shows how we can address this problem in a plain recurrent network by analyzing the gating mechanisms in GNNs. We propose a novel network called the Recurrent Identity Network (RIN) which allows a plain recurrent network to overcome the vanishing gradient problem while training very deep models without the use of gates. We compare this model with IRNNs and LSTMs on multiple sequence modeling benchmarks. The RINs demonstrate competitive performance and converge faster in all tasks. Notably, small RIN models produce 12%–67% higher accuracy on the Sequential and Permuted MNIST datasets and reach state-of-the-art performance on the bAbI question answering dataset.

## 1 Introduction

Numerous methods have been proposed for mitigating the vanishing gradient problem including the use of second-order optimization methods (*e.g.*, Hessian-free optimization (Martens & Sutskever, 2011)), specific training schedules (*e.g.*, Greedy Layer-wise training (Schmidhuber, 1992; Hinton et al., 2006; Vincent et al., 2008)), and special weight initialization methods when training on both plain FFNs and RNNs (Glorot & Bengio, 2010; Mishkin & Matas, 2015; Le et al., 2015; Jing et al., 2016; Xie et al., 2017; Jing et al., 2017).

Gated Neural Networks (GNNs) also help to mitigate this problem by introducing "gates" to control information flow through the network over layers or sequences. Notable examples include recurrent networks such as Long-short Term Memory (LSTM) (Hochreiter & Schmidhuber, 1997), Gated Recurrent Unit (GRU) (Chung et al., 2014; Cho et al., 2014), and feedforward networks such as Highway Networks (HNs) (Srivastava et al., 2015), and Residual Networks (ResNets) (He et al., 2015). One can successfully train very deep models by employing these models, *e.g.*, ResNets can be trained with over 1,000 layers. It has been demonstrated that removing (lesioning) or reordering (re-shuffling) random layers in deep feedforward GNNs does not noticeable affect the performance of the network (Veit et al., 2016) Noticeably, one interpretation for this effect as given by Greff et al. (2016) is that the functional blocks in HNs or ResNets engage in an Unrolled Iterative Estimate (UIE) of representations and that layers in this block of HNs or ResNets iteratively refine a single set of representations.

In this paper, we investigate if the view of *Iterative Estimation* (IE) can also be applied towards recurrent GNNs (Section 2.1). We present a formal analysis for GNNs by examining a dual gate design common in LSTM and GRU (Section 2.2). The analysis suggests that the use of gates in GNNs encourages the network to learn an identity mapping which can be beneficial in training deep architectures (He et al., 2016; Greff et al., 2016).

We propose a new formulation of a plain RNN, called a Recurrent Identity Network (RIN), that is encouraged to learn an identity mapping without the use of gates (Section 2). This network uses ReLU as the activation function and contains a set of non-trainable parameters. This simple yet effective method helps the plain recurrent network to overcome the vanishing gradient problem while it is still able to model long-range dependencies. This network is compared against two competing networks, the IRNN (Le et al., 2015) and LSTM, on several long sequence modeling tasks including

the adding problem (Section 3.1), Sequential and Permuted MNIST classification tasks (Section 3.2), and bAbI question answering tasks (Section 3.3). RINs show faster convergence than IRNNs and LSTMs in the early stage of the training phase and reach competitive performance in all benchmarks. Note that the use of ReLU in RNNs usually leads to training instability, and therefore the network is sensitive to training hyperparameters. Our proposed RIN network demonstrates that a plain RNN does not suffer from this problem even with the use of ReLUs as shown in Section 3. We discuss further implications of this network and related work in Section 4.

## 2 METHODS

### 2.1 ITERATIVE ESTIMATION VIEW IN RNNS

Representation learning in RNNs requires that the network build a latent state, which reflects the temporal dependencies over a sequence of inputs. In this section, we explore an interpretation of this process using iterative estimation (IE), a view that is similar to the UIE view for feedforward GNNs. Formally, we characterize this viewpoint in Eq. 1, that is, the expectation of the difference between the hidden activation at step $t$, $\mathbf{h}_t$, and the last hidden activation at step $T$, $\mathbf{h}_T$, is zero:

$$\mathbb{E}_{\mathbf{x}_{1,\dots,T}}\left[\mathbf{h}_t - \mathbf{h}_T\right] = 0. \tag{1}$$

This formulation implies that an RNN layer maintains and updates the same set of representations over the input sequence. Given the fact that the hidden activation at every step is an estimation of the final activation, we derive Eq. 3.

$$\mathbb{E}_{\mathbf{x}_{1,\dots,T}}\left[\mathbf{h}_t - \mathbf{h}_T\right] - \mathbb{E}_{\mathbf{x}_{1,\dots,T}}\left[\mathbf{h}_{t-1} - \mathbf{h}_T\right] = 0 \tag{2}$$

$$\Rightarrow \quad \mathbb{E}_{\mathbf{x}_{1,\dots,T}}\left[\mathbf{h}_t - \mathbf{h}_{t-1}\right] = 0 \tag{3}$$

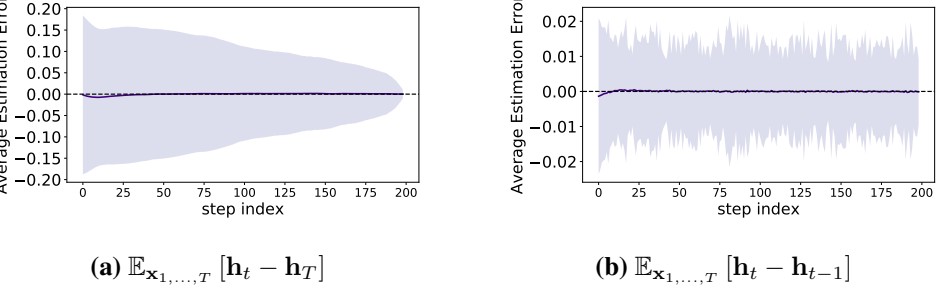

**(a)** $\mathbb{E}_{\mathbf{x}_{1,\dots,T}}\left[\mathbf{h}_t - \mathbf{h}_T\right]$  **(b)** $\mathbb{E}_{\mathbf{x}_{1,\dots,T}}\left[\mathbf{h}_t - \mathbf{h}_{t-1}\right]$

Figure 1: Observation of learning identity mapping in an LSTM model trained on the adding problem task (see Section 3.1). The average estimation error is computed over a batch of 128 samples of the test set. **(a)** and **(b)** show the evaluation of Eq. 1 and Eq. 3 respectively. The x-axis indicates the index of the step that compares with the final output $\mathbf{h}_T$ or its previous step $\mathbf{h}_{t-1}$.

Fig. 1 shows an empirical observation of the IE in the adding problem (experimental details in Section 3.1). Here, we use the Average Estimation Error (AEE) measure (Greff et al., 2016) to quantify the expectation of the difference between two hidden activations. The measured AEEs in Fig. 1 are close to 0 indicating that the LSTM model fulfills the view of IE. The results also suggest that the network learns an identity mapping since the activation levels are similar on average across all recurrent updates. In the next section, we shall show that the use of gates in GNNs encourages the network to learn an identity mapping and whether this analysis can be extended to plain recurrent networks.

### 2.2 ANALYSIS OF GNNS

Popular GNNs such as LSTM, GRU; and recent variants such as the Phased-LSTM (Neil et al., 2016), and Intersection RNN (Collins et al., 2017), share the same dual gate design following:

$$\mathbf{h}_t = \mathbf{H}_t \odot \mathbf{T}_t + \mathbf{h}_{t-1} \odot \mathbf{C}_t \tag{4}$$

where $t \in [1, T]$, $\mathbf{H}_t = \sigma(\mathbf{x}_t, \mathbf{h}_{t-1})$ represents the hidden transformation, $\mathbf{T}_t = \tau(\mathbf{x}_t, \mathbf{h}_{t-1})$ is the transform gate, and $\mathbf{C}_t = \phi(\mathbf{x}_t, \mathbf{h}_{t-1})$ is the carry gate. $\sigma$, $\tau$ and $\phi$ are recurrent layers that have their trainable parameters and activation functions. $\odot$ represents element-wise product operator. Note that $\mathbf{h}_t$ may not be the output activation at the recurrent step $t$. For example in LSTM, $\mathbf{h}_t$ represents the memory cell state. Typically, the elements of transform gate $\mathbf{T}_{t,k}$ and carry gate $\mathbf{C}_{t,k}$ are between 0 (close) and 1 (open), the value indicates the openness of the gate at the $k$th neuron. Hence, a plain recurrent network is a subcase of Eq. 4 when $\mathbf{T}_t = \mathbf{1}$ and $\mathbf{C}_t = \mathbf{0}$.

Note that conventionally, the initial hidden activation $\mathbf{h}_0$ is $\mathbf{0}$ to represent a "void state" at the start of computation. For $\mathbf{h}_0$ to fit into Eq. 4's framework, we define an auxiliary state $\mathbf{h}_{-1}$ as the previous state of $\mathbf{h}_0$, and $\mathbf{T}_0 = \mathbf{1}$, $\mathbf{C}_0 = \mathbf{0}$. We also define another auxiliary state $\mathbf{h}_{T+1} = \mathbf{h}_T$, $\mathbf{T}_{T+1} = \mathbf{0}$, and $\mathbf{C}_{T+1} = \mathbf{1}$ as the succeeding state of $\mathbf{h}_T$.

Based on the recursive definition in Eq. 4, we can write the final layer output $\mathbf{h}_T$ as follows:

$$\mathbf{h}_T = \mathbf{h}_0 \odot \prod_{t=1}^{T} \mathbf{C}_t + \sum_{t=1}^{T} \left( \mathbf{H}_t \odot \mathbf{T}_t \odot \prod_{i=t+1}^{T+1} \mathbf{C}_i \right) \tag{5}$$

where we use $\prod$ to represent element-wise multiplication over a series of terms.

According to Eq. 3, and supposing that Eq. 5 fulfills the Eq. 1, we can use a zero-mean residual $\boldsymbol{\epsilon}_t$ for describing the difference between the outputs of recurrent steps:

$$\mathbf{h}_t - \mathbf{h}_{t-1} = \boldsymbol{\epsilon}_t \tag{6}$$
$$\boldsymbol{\epsilon}_0 = \mathbf{0} \tag{7}$$

Plugging Eq. 6 into Eq. 5, we get

$$\mathbf{h}_T = \mathbf{h}_0 + \boldsymbol{\lambda} \tag{8}$$

where

$$\boldsymbol{\lambda} = \sum_{t=1}^{T} \boldsymbol{\lambda}_t = \sum_{t=1}^{T} \left( \left( \sum_{i=1}^{t} \boldsymbol{\epsilon}_i \right) \odot \prod_{j=t+1}^{T+1} \mathbf{C}_j - \left( \sum_{i=0}^{t-1} \boldsymbol{\epsilon}_i \right) \odot \prod_{j=t}^{T} \mathbf{C}_j \right) \tag{9}$$

The complete deduction of Eqs. 8–9 is presented in Appendix A. Eq. 8 performs an *identity mapping* when the carry gate $\mathbf{C}_t$ is always open. In Eq. 9, the term $\sum_{i=1}^{t} \boldsymbol{\epsilon}_i$ represents "a level of representation that is formed between $\mathbf{h}_1$ and $\mathbf{h}_t$". Moreover, the term $\prod_{j=t}^{T} \mathbf{C}_j$ extract the "useful" part of this representation and contribute to the final representation of the recurrent layer. Here, we interpret "useful" as any quantity that helps in minimizing the cost function. Therefore, the contribution, $\boldsymbol{\lambda}_t$, at each recurrent step, quantifies the representation that is learned in the step $t$. Furthermore, it is generally believed that a GNN manages and maintains the latent state through the carry gate, such as the forget gate in LSTM. If the carry gate is closed, then it is impossible for the old state to be preserved while undergoing recurrent updates. However, if we set $\mathbf{C}_t = \mathbf{0}$, $t \in [1, T]$ in Eq. 9, we get:

$$\mathbf{h}_T = \mathbf{h}_0 + \sum_{t=1}^{T} \boldsymbol{\epsilon}_t \tag{10}$$

If $\mathbf{h}_0 = \mathbf{0}$ (void state at the start), we can turn Eq. 10 into:

$$\mathbf{h}_T = \boldsymbol{\epsilon}_1 + \sum_{t=2}^{T} \boldsymbol{\epsilon}_t = \mathbf{h}_1 + \sum_{t=2}^{T} \boldsymbol{\epsilon}_t \tag{11}$$

Eq. 11 shows that the state can be preserved without the help of the carry gate. This result indicates that it is possible for a plain recurrent network to learn an identity mapping as well.

## 2.3 RECURRENT IDENTITY NETWORKS

Motivated by the previous iterative estimation interpretation of RNNs, we formulate a novel plain recurrent network variant — Recurrent Identity Network (RIN):

$$\mathbf{h}_t = \text{ReLU} \left( \mathbf{W}\mathbf{x}_t + \mathbf{U}\mathbf{h}_{t-1} + \mathbf{h}_{t-1} + \mathbf{b} \right) \tag{12}$$
$$= \text{ReLU} \left( \mathbf{W}\mathbf{x}_t + (\mathbf{U} + \mathbf{I})\mathbf{h}_{t-1} + \mathbf{b} \right) \tag{13}$$

where $\mathbf{W}$ is the input-to-hidden weight matrix, $\mathbf{U}$ is the hidden-to-hidden weight matrix, and $\mathbf{I}$ is a non-trainable identity matrix that acts as a "surrogate memory" component. This formulation encourages the network to preserve a copy of the last state by embedding $\mathbf{I}$ into the hidden-to-hidden weights. This "surrogate memory" component maintains the representation encoded in the past recurrent steps.

## 3   RESULTS

In this section, we compare the performances of the RIN, IRNN, and LSTM in a set of tasks that require modeling long-range dependencies.

### 3.1   THE ADDING PROBLEM

The adding problem is a standard task for examining the capability of RNNs for modeling long-range dependencies (Hochreiter & Schmidhuber, 1997). In this task, two numbers are randomly selected from a long sequence. The network has to predict the sum of these two numbers. The task becomes challenging as the length of the sequence $T$ increases because the relevant numbers can be far from each other in a long sequence. We report experimental results from three datasets that have sequence lengths of $T_1 = 200$, $T_2 = 300$, and $T_3 = 400$ respectively. Each dataset has 100,000 training samples and 10,000 testing samples. Each sequence of a dataset has $T_i$ numbers that are randomly sampled from a uniform distribution in $[0, 1]$. Each sequence is accompanied by a mask that indicates the two chosen random positions.

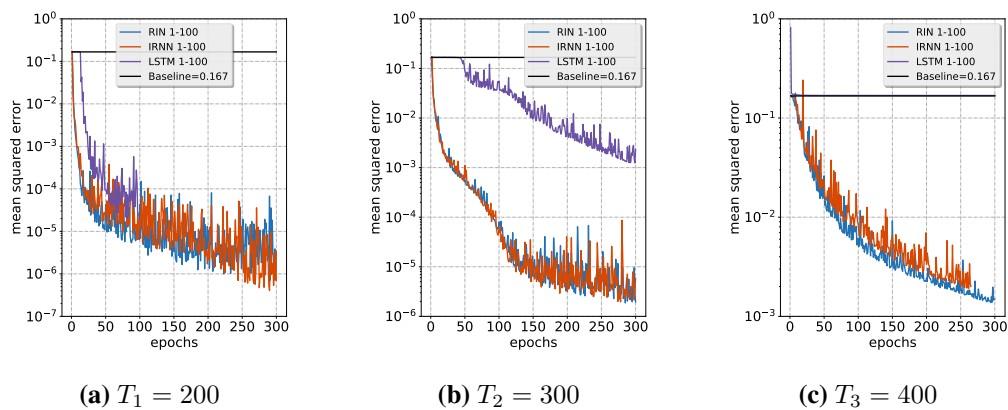

**(a)** $T_1 = 200$ **(b)** $T_2 = 300$ **(c)** $T_3 = 400$

Figure 2: Mean Squared Error (MSE) plots during the testing phase of the adding problem task for different sequence lengths. The errors are presented in log scale. LSTM performs the worst in all three tasks. RIN and IRNN models are comparable in **(a)** and **(b)**, and **(c)**.

We compare the performance between RINs, IRNNs, and LSTMs using the same experimental settings. Each network has one hidden layer with 100 hidden units. Note that a LSTM has four times more parameters than corresponding RIN and IRNN models. The optimizer minimizes the *Mean Squared Error* (MSE) between the target sum and the predicted sum. We initially used the RMSprop (Tieleman & Hinton, 2012) optimizer. However, some IRNN models failed to converge using this optimizer. Therefore, we chose the Adam optimizer (Kingma & Ba, 2014) so a fair comparison can be made between the different networks. The batch size is 32. Gradient clipping value for all models is 100. The models are trained with maximum 300 epochs until they converged. The initial learning rates are different between the datasets because we found that IRNNs are sensitive to the initial learning rate as the sequence length increases. The learning rates $\alpha_{200} = 10^{-4}$, $\alpha_{300} = 10^{-5}$ and $\alpha_{400} = 10^{-6}$ are applied to $T_1$, $T_2$ and $T_3$ correspondingly. The input-to-hidden weights of RINs and IRNNs and hidden-to-hidden weights of RINs are initialized using a similar method to Le et al. (2015) where the weights are drawn from a Gaussian distribution $\mathcal{N}(0, 10^{-3})$. The LSTM is initialized with the settings where the input-to-hidden weights use Glorot Uniform (Glorot & Bengio, 2010) and hidden-to-hidden weights use an orthogonal matrix as suggested by Saxe et al.

(2013). Bias values for all networks are initialized to 0. No explicit regularization is employed. We do not perform an exhaustive hyperparameter search in these experiments.

The baseline MSE of the task is 0.167. This score is achieved by predicting the sum of two numbers as 1 regardless of the input sequence. Fig. 2 shows MSE plots for different test datasets. RINs and IRNNs reached the same level of performance in all experiments, and LSTMs performed the worst. Notably, LSTM fails to converge in the dataset with $T_3 = 400$. The use of ReLU in RINs and IRNNs causes some degree of instability in the training phase. However, in most cases, RINs converge faster and are more stable than IRNNs (see training loss plots in Fig. 5 of Appendix B). Note that because IRNNs are sensitive to the initial learning rate, applying high learning rates such as $\alpha = 10^{-3}$ for $T_2$ and $T_3$ could cause the training of the network to fail.

## 3.2 SEQUENTIAL AND PERMUTED MNIST

Sequential and Permuted MNIST are introduced by Le et al. (2015) for evaluating RNNs. **Sequential MNIST** presents each pixel of the MNIST handwritten image (Lecun et al., 1998) to the network sequentially (*e.g.*, from the top left corner of the image to the bottom right corner of the image). After the network has seen all $28 \times 28 = 784$ pixels, the network produces the class of the image. This task requires the network to model a very long sequence that has 784 steps. **Permuted MNIST** is an even harder task than the Sequential MNIST in that a fixed random index permutation is applied to all images. This random permutation breaks the association between adjacent pixels. The network is expected to find the hidden relations between pixels so that it can correctly classify the image.

All networks are trained with the RMSprop optimizer (Tieleman & Hinton, 2012) and a batch size of 128. The networks are trained with maximum 500 epochs until they are converged. The initial learning rate is set to $\alpha = 10^{-6}$. Weight initialization follows the same setup as Section 3.1. No explicit regularization is added.

Table 1 summarizes the accuracy performance of the networks on the Sequential and Permuted MNIST datasets. For small network sizes (1–100, 1–200), RINs outperform IRNNs in their accuracy performance. For bigger networks, RINs and IRNNs achieve similar performance; however, RINs converge much faster than IRNNs in the early stage of training (see Fig. 3). LSTMs perform the worst on both tasks in terms of both convergence speed and final accuracy. Appendix C presents the full experimental results.

To investigate the limit of RINs, we adopted the concept of Deep Transition (DT) Networks (Pascanu et al., 2013) for increasing the implicit network depth. In this extended RIN model called RIN-DT, each recurrent step performs two hidden transitions instead of one (the formulation is given in Appendix D). The network modification increases the inherent depth by a factor of two. The results showed that the error signal could survive $784 \times 2 = 1568$ computation steps in RIN-DTs.

In Fig. 4, we show the evidence of learning identity mapping empirically by collecting the hidden activation from all recurrent steps and evaluating Eqs. 1 and 3. The network matches the IE when AEE is close to zero. We also compute the variance of the difference between two recurrent steps. Fig. 4(a) suggests that all networks bound the variance across recurrent steps. Fig. 4(b) offers a closer perspective where it measures the AEE between two adjacent steps. The levels of activations for all networks are always kept the same on an average, which is an evidence of learning identity mapping. We also observed that the magnitude of the variance becomes significantly larger at the last 200 steps in IRNN and RIN. Repeated application of ReLU may cause this effect during recurrent update (Jastrzebski et al., 2017). Other experiments in this section exhibit similar behaviors, complete results are shown in Appendix C (Fig. 8–12). Note that this empirical analysis only demonstrates that the tested RNNs have the evidence of learning identity mapping across recurrent updates as RINs and IRNNs largely fulfill the view of IE. We do not over-explain the relationship between this analysis and the performance of the network.

Table 1: Accuracies of RINs, IRNNs and LSTM on Sequential and Permuted MNIST. The network type is represented by `No. layers-No. units`, *e.g.*, 3–100 means that the network has 3 layers and each layer has 100 hidden units. The LSTM results matches with Le et al. (2015)

| Network Type | Sequential MNIST | | | Permuted MNIST | | |
|---|---|---|---|---|---|---|
| | **RIN** | **IRNN** | **LSTM** | **RIN** | **IRNN** | **LSTM** |
| 1–100 | 91.64% | 83.55% | 24.10% | 78.89% | 62.11% | 28.49% |
| 1–200 | 94.60% | 92.86% | 47.13% | 85.03% | 73.73% | 30.63% |
| 2–100 | 93.69% | 92.15% | 39.50% | 83.37% | 76.31% | 41.31% |
| 2–200 | 94.82% | 94.78% | 22.27% | 85.31% | 83.78% | 55.44% |
| 3–100 | 94.15% | 94.03% | 54.98% | 84.15% | 78.78% | 38.61% |
| 3–200 | 95.19% | 95.05% | 61.20% | 83.41% | 84.24% | 53.29% |
| | **RIN-DT** | | | **RIN-DT** | | |
| 1–100 | **95.41%** | | | **86.23%** | | |

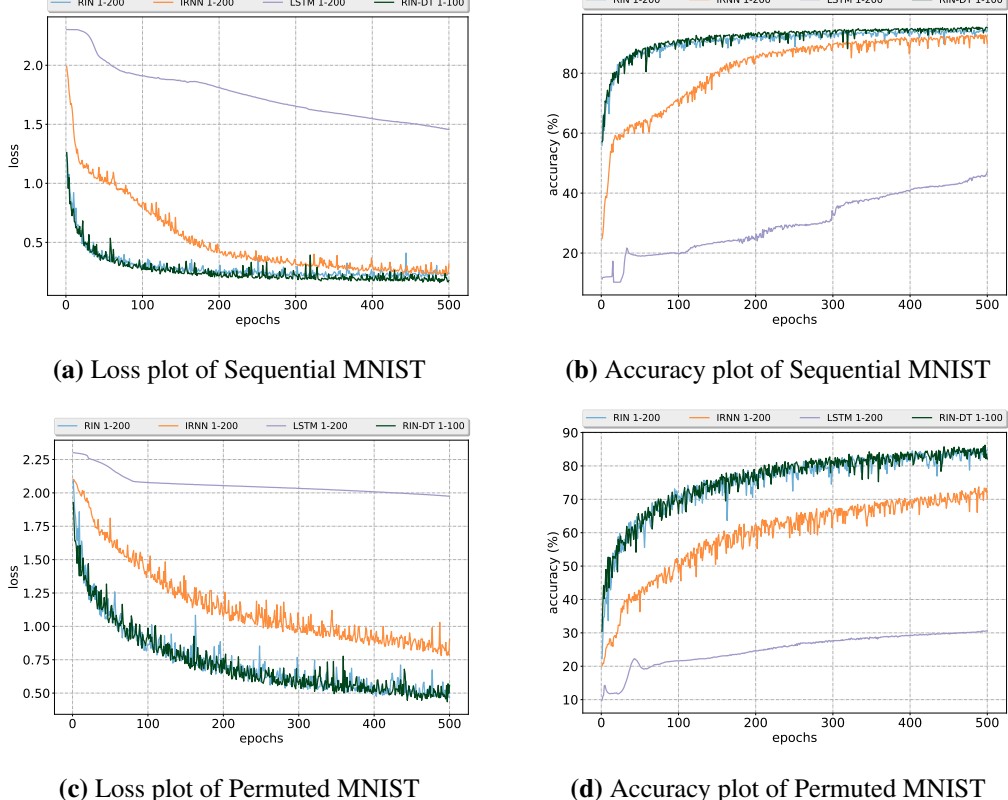

**(a)** Loss plot of Sequential MNIST

**(b)** Accuracy plot of Sequential MNIST

**(c)** Loss plot of Permuted MNIST

**(d)** Accuracy plot of Permuted MNIST

Figure 3: Results for network type 1–200. **(a)** and **(b)** show the loss and accuracy curves on Sequential MNIST; **(c)** and **(d)** present the loss and accuracy curves on Permuted MNIST. RINs and RIN-DTs converge much faster than IRNNs and LSTMs in the early stage of training (first 100 epochs) and achieve a better final accuracy.

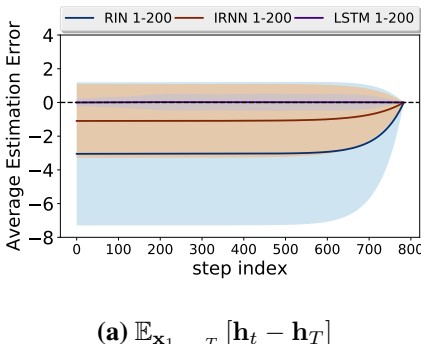 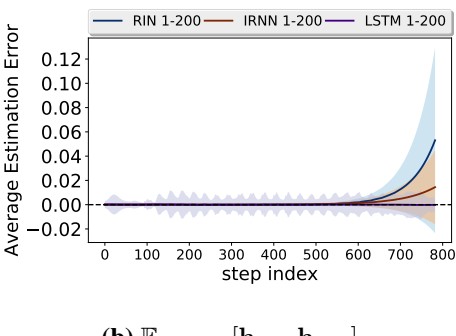

**(a)** $\mathbb{E}_{\mathbf{x}_{1,\ldots,T}}\left[\mathbf{h}_t - \mathbf{h}_T\right]$          **(b)** $\mathbb{E}_{\mathbf{x}_{1,\ldots,T}}\left[\mathbf{h}_t - \mathbf{h}_{t-1}\right]$

Figure 4: Evidence of learning identity mapping in RIN, IRNN and LSTM for network type 1–200 over a batch of 128 samples. **(a)** evaluates Eq. 1 and **(b)** evaluates Eq. 3. The x-axis indicates the index of the step that compares with the final output $\mathbf{h}_T$ or its previous step $\mathbf{h}_{t-1}$, and y-axis represents the average estimation error (AEE).

### 3.3 BABI QUESTION ANSWERING TASKS

The bAbI dataset provides 20 question answering tasks that measure the understanding of language and the performance of reasoning in neural networks (Weston et al., 2015). Each task consists of 1,000 training samples and 1,000 test samples. A sample consists of three parts: a list of statements, a question and an answer (examples in Table 2). The answer to the question can be inferred from the statements that are logically organized together.

Table 2: Examples of bAbI tasks.

| Statements: | | Statements: | |
|---|---|---|---|
| Mary went to the office. | | The red square is below the blue square. | |
| Then she journeyed to the garden. | | The red square is to the left of the pink rectangle. | |
| **Question:** | Where is Mary? | **Question:** | Is the blue square below the pink rectangle? |
| **Answer:** | Garden. | **Answer:** | No. |

We compare the performance of the RIN, IRNN, and LSTM on these tasks. All networks follow a network design where the network firstly embeds each word into a vector of 200 dimensions. The statements are then appended together to a single sequence and encoded by a recurrent layer while another recurrent layer encodes the question sequence. The outputs of these two recurrent layers are concatenated together, and this concatenated sequence is then passed to a different recurrent layer for decoding the answer. Finally, the network predicts the answer via a softmax layer. The recurrent layers in all networks have 100 hidden units. This network design roughly follows the architecture presented in Jing et al. (2017). The initial learning rates are set to $\alpha = 10^{-3}$ for RINs and LSTMs and $\alpha = 10^{-4}$ for IRNNs because IRNNs fail to converge with a higher learning rate on many tasks. We chose the Adam optimizer over the RMSprop optimizer because of the same reasons as in the adding problem. The batch size is 32. Each network is trained for maximum 100 epochs until the network converges. The recurrent layers in the network follow the same initialization steps as in Section 3.1.

The results in Table 3 show that RINs can reach mean performance similar to the state-of-the-art performance reported in Jing et al. (2017). As discussed in Section 3.1, the use of ReLU as the activation function can lead to instability during training of IRNN for tasks that have lengthy statements (*e.g.*. 3-Three Supporting Facts, 5-Three Arg. Relations).

Table 3: Test accuracy (%) of 20 bAbI Question Answering Tasks.

| Task | RIN | IRNN | LSTM | Jing et al. (2017) | Weston et al. (2015) |
|---|---|---|---|---|---|
| 1: Single Supporting Fact | 51.9 | 48.4 | 50.3 | 48.2 | 50 |
| 2: Two Supporting Facts | 18.7 | 18.7 | 19 | 15.8 | 20 |
| 3: Three Supporting Facts | 18.5 | 15.3 | 22.9 | 19.1 | 20 |
| 4: Two Arg. Relations | 71.2 | 72.6 | 71.6 | 75.8 | 61 |
| 5: Three Arg. Relations | 16.4 | 18.9 | 36.4 | 33.7 | 70 |
| 6: Yes/No Questions | 50.3 | 50.3 | 52.3 | 49 | 48 |
| 7: Counting | 48.8 | 48.8 | 48.9 | 48 | 49 |
| 8: Lists/Sets | 33.6 | 33.6 | 33.6 | 33.6 | 45 |
| 9: Simple Negation | 64.6 | 64.7 | 63.8 | 63.2 | 64 |
| 10: Indefinite Knowledge | 45.1 | 43.7 | 45.1 | 43.9 | 44 |
| 11: Basic Coreference | 71.6 | 67.8 | 78.4 | 68.8 | 72 |
| 12: Conjunction | 70.6 | 71.4 | 75.3 | 73 | 74 |
| 13: Compound Coref. | 94.4 | 94.2 | 94.4 | 93.9 | 94 |
| 14: Time Reasoning | 36.7 | 17.6 | 23.2 | 19.7 | 27 |
| 15: Basic Deduction | 54.8 | 54.1 | 26.7 | 54.9 | 21 |
| 16: Basic Induction | 48.8 | 49 | 25.8 | 46.6 | 23 |
| 17: Positional Reasoning | 53.9 | 53.4 | 52 | 60.5 | 51 |
| 18: Size Reasoning | 92.6 | 46.9 | 93 | 91.3 | 52 |
| 19: Path Finding | 10.5 | 10.9 | 9.9 | 10 | 8 |
| 20: Agent's Motivations | 98 | 98.2 | 97.3 | 97.4 | 91 |
| Mean Performance | **52.6** | 48.9 | 51.0 | 52.3 | 49.2 |

## 4 DISCUSSION

In this paper, we discussed the iterative representation refinement in RNNs and how this viewpoint could help in learning identity mapping. Under this observation, we demonstrated that the contribution of each recurrent step a GNN can be jointly determined by the representation that is formed up to the current step, and the openness of the carry gate in later recurrent updates. Note in Eq. 9, the element-wise multiplication of $C_t$s selects the encoded representation that could arrive at the output of the layer. Thus, it is possible to embed a special function in $C_t$s so that they are sensitive to certain pattern of interests. For example, in Phased LSTM, the time gate is inherently interested in temporal frequency selection (Neil et al., 2016).

Motivated by the analysis presented in Section 2, we propose a novel plain recurrent network variant, the Recurrent Identity Network (RIN), that can model long-range dependencies without the use of gates. Compared to the conventional formulation of plain RNNs, the formulation of RINs only adds a set of non-trainable weights to represent a "surrogate memory" component so that the learned representation can be maintained across two recurrent steps.

Experimental results in Section 3 show that RINs are competitive against other network models such as IRNNs and LSTMs. Particularly, small RINs produce 12%–67% higher accuracy in the Sequential and Permuted MNIST. Furthermore, RINs demonstrated much faster convergence speed in early phase of training, which is a desirable advantage for platforms with limited computing resources. RINs work well without advanced methods of weight initializations and are relatively insensitive to hyperparameters such as learning rate, batch size, and selection of optimizer. This property can be very helpful when the time available for choosing hyperparameters is limited. Note that we do not claim that RINs outperform LSTMs in general because LSTMs may achieve comparable performance with finely-tuned hyperparameters.

The use of ReLU in RNNs might be counterintuitive at first sight because the repeated application of this activation is more likely causing gradient explosion than conventional choices of activation function, such as hyperbolic tangent (tanh) function or sigmoid function. Although the proposed IRNN (Le et al., 2015) reduces the problem by the identity initialization, in our experiments, we usually found that IRNN is more sensitive to training parameters and more unstable than RINs and LSTMs. On the contrary, feedforward models that use ReLU usually produce better results and converge faster than FFNs that use the tanh or sigmoid activation function. In this paper, we provide a promising method of using ReLU in RNNs so that the network is less sensitive to the training

conditions. The experimental results also support the argument that the use of ReLU significantly speeds up the convergence.

During the development of this paper, a recent independent work (Zagoruyko & Komodakis, 2017) presented a similar network formulation with a focus on training of deep plain FFNs without skip connections. DiracNet uses the idea of ResNets where it assumes that the identity initialization can replace the role of the skip-connection in ResNets. DiracNet employed a particular kind of activation function — negative concatenated ReLU (NCReLU), and this activation function allows the layer output to approximate the layer input when the expectation of the weights are close to zero. In this paper, we showed that an RNN can be trained without the use of gates or special activation functions, which complements the findings and provides theoretical basis in Zagoruyko & Komodakis (2017).

We hope to see more empirical and theoretical insights that explains the effectiveness of the RIN by simply embedding a non-trainable identity matrix. In future, we will investigate the reasons for the faster convergence speed of the RIN during training. Furthermore, we will investigate why RIN can be trained stably with the repeated application of ReLU and why it is less sensitive to training parameters than the two other models.

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

## A ALGEBRA OF EQS. 8–9

Popular GNNs such as LSTM, GRU; and recent variants such as the Phased-LSTM (Neil et al., 2016), and Intersection RNN (Collins et al., 2017), share the same dual gate design described as follows:

$$\mathbf{h}_t = \mathbf{H}_t \odot \mathbf{T}_t + \mathbf{h}_{t-1} \odot \mathbf{C}_t \tag{14}$$

where $t \in [1, T]$, $\mathbf{H}_t = \sigma(\mathbf{x}_t, \mathbf{h}_{t-1})$ represents the hidden transformation, $\mathbf{T}_t = \tau(\mathbf{x}_t, \mathbf{h}_{t-1})$ is the transform gate, and $\mathbf{C}_t = \phi(\mathbf{x}_t, \mathbf{h}_{t-1})$ is the carry gate. $\sigma$, $\tau$ and $\phi$ are recurrent layers that have their trainable parameters and activation functions. $\odot$ represents element-wise product operator. Note that $\mathbf{h}_t$ may not be the output activation at the recurrent step $t$. For example in LSTM, $\mathbf{h}_t$ represents the memory cell state. Typically, the elements of transform gate $\mathbf{T}_{t,k}$ and carry gate $\mathbf{C}_{t,k}$ are between 0 (close) and 1 (open), the value indicates the openness of the gate at the $k$th neuron. Hence, a plain recurrent network is a subcase of Eq. 14 when $\mathbf{T}_t = \mathbf{1}$ and $\mathbf{C}_t = \mathbf{0}$.

Note that conventionally, the initial hidden activation $\mathbf{h}_0$ is $\mathbf{0}$ to represent a "void state" at the start of computation. For $\mathbf{h}_0$ to fit into Eq. 4's framework, we define an auxiliary state $\mathbf{h}_{-1}$ as the previous state of $\mathbf{h}_0$, and $\mathbf{T}_0 = \mathbf{1}$, $\mathbf{C}_0 = \mathbf{0}$. We also define another auxiliary state $\mathbf{h}_{T+1} = \mathbf{h}_T$, $\mathbf{T}_{T+1} = \mathbf{0}$, and $\mathbf{C}_{T+1} = \mathbf{1}$ as the succeeding state of $\mathbf{h}_T$.

Based on the recursive definition in Eq. 4, we can write the final layer output $\mathbf{h}_T$ as follows:

$$\mathbf{h}_T = \mathbf{h}_0 \odot \prod_{t=1}^{T} \mathbf{C}_t + \sum_{t=1}^{T} \left( \mathbf{H}_t \odot \mathbf{T}_t \odot \prod_{i=t+1}^{T+1} \mathbf{C}_i \right) \tag{15}$$

where we use $\prod$ to represent element-wise multiplication over a series of terms.

According to Eq. 3, and supposing that Eq. 5 fulfills the Eq. 1, we can use a zero-mean residual $\boldsymbol{\epsilon}_t$ for describing the difference between the outputs of recurrent steps:

$$\mathbf{h}_t - \mathbf{h}_{t-1} = \boldsymbol{\epsilon}_t \tag{16}$$

$$\boldsymbol{\epsilon}_0 = \mathbf{0} \tag{17}$$

Then we can rewrite Eq. 16 as:

$$\mathbf{H}_t \odot \mathbf{T}_t + \mathbf{h}_{t-1} \odot \mathbf{C}_t = \mathbf{h}_{t-1} + \boldsymbol{\epsilon}_t \tag{18}$$

Substituting Eq. 18 into Eq. 15:

$$\mathbf{h}_T = \mathbf{h}_0 \odot \prod_{t=1}^{T} \mathbf{C}_t + \sum_{t=1}^{T} \left( ((1 - \mathbf{C}_t) \odot \mathbf{h}_{t-1} + \boldsymbol{\epsilon}_t) \odot \prod_{j=t+1}^{T+1} \mathbf{C}_j \right) \tag{19}$$

$$= \mathbf{h}_0 \odot \prod_{t=1}^{T} \mathbf{C}_t + \sum_{t=1}^{T} \left( \left( (1 - \mathbf{C}_t) \odot \left( \mathbf{h}_0 + \sum_{i=1}^{t-1} \boldsymbol{\epsilon}_i \right) + \boldsymbol{\epsilon}_t \right) \odot \prod_{j=t+1}^{T+1} \mathbf{C}_j \right) \tag{20}$$

We can rearrange Eqn. 20 to

$$\mathbf{h}_T = \mathbf{h}_0 \odot \left( \sum_{t=0}^{T} (\mathbf{1} - \mathbf{C}_t) \prod_{i=t+1}^{T+1} \mathbf{C}_i \right) + \boldsymbol{\lambda} \tag{21}$$

$$= \mathbf{h}_0 \odot \left( \mathbf{C}_{T+1} - \prod_{t=0}^{T+1} \mathbf{C}_t \right) + \boldsymbol{\lambda} \tag{22}$$

$$= \mathbf{h}_0 + \boldsymbol{\lambda} \tag{Eq. 8}$$

where

$$\boldsymbol{\lambda} = \sum_{t=1}^{T} \boldsymbol{\lambda}_t = \sum_{t=1}^{T} \left( \left( (1 - \mathbf{C}_t) \odot \sum_{i=1}^{t-1} \boldsymbol{\epsilon}_i + \boldsymbol{\epsilon}_t \right) \odot \prod_{j=t+1}^{T+1} \mathbf{C}_j \right) \tag{23}$$

The term $\boldsymbol{\lambda}$ in Eq. 23 can be reorganized to,

$$\boldsymbol{\lambda} = \sum_{t=1}^{T} \boldsymbol{\lambda}_t = \sum_{t=1}^{T} \left( \left( (\mathbf{1} - \mathbf{C}_t) \odot \sum_{i=1}^{t-1} \boldsymbol{\epsilon}_i + \boldsymbol{\epsilon}_t \right) \odot \prod_{j=t+1}^{T+1} \mathbf{C}_j \right) \tag{24}$$

$$= \sum_{t=1}^{T} \left( \left( \sum_{i=1}^{t} \boldsymbol{\epsilon}_i - \mathbf{C}_t \odot \sum_{i=0}^{t-1} \boldsymbol{\epsilon}_i \right) \odot \prod_{j=t+1}^{T+1} \mathbf{C}_j \right) \tag{25}$$

$$= \sum_{t=1}^{T} \left( \sum_{i=1}^{t} \boldsymbol{\epsilon}_i \odot \prod_{j=t+1}^{T+1} \mathbf{C}_j - \sum_{i=0}^{t-1} \boldsymbol{\epsilon}_i \odot \prod_{j=t}^{T} \mathbf{C}_j \right) \tag{Eq. 9}$$

# B  DETAILS IN THE ADDING PROBLEM EXPERIMENTS

Fig. 5 presents the MSE plots during the training phase. As we discussed in Section 3.1, the choice of ReLU can occur some degree of instability during the training. Compared to RINs, IRNNs are much more unstable in $T_3 = 400$.

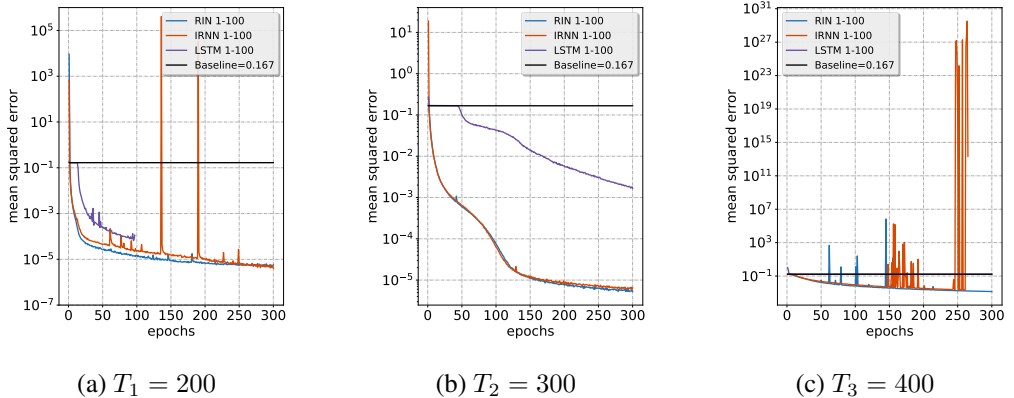

(a) $T_1 = 200$        (b) $T_2 = 300$        (c) $T_3 = 400$

Figure 5: Mean Squared Error (MSE) plots for the adding problem with different sequence lengths at training phase. The figures are presented in log scale. All models are trained up to 300 epochs.

## C    DETAILS IN SEQUENTIAL AND PERMUTED MNIST EXPERIMENTS

Fig. 6–7 show all training and testing curves for Sequential and Permuted MNIST experiments. RINs and RIN-DTs converge much faster than IRNNs and LSTMs.

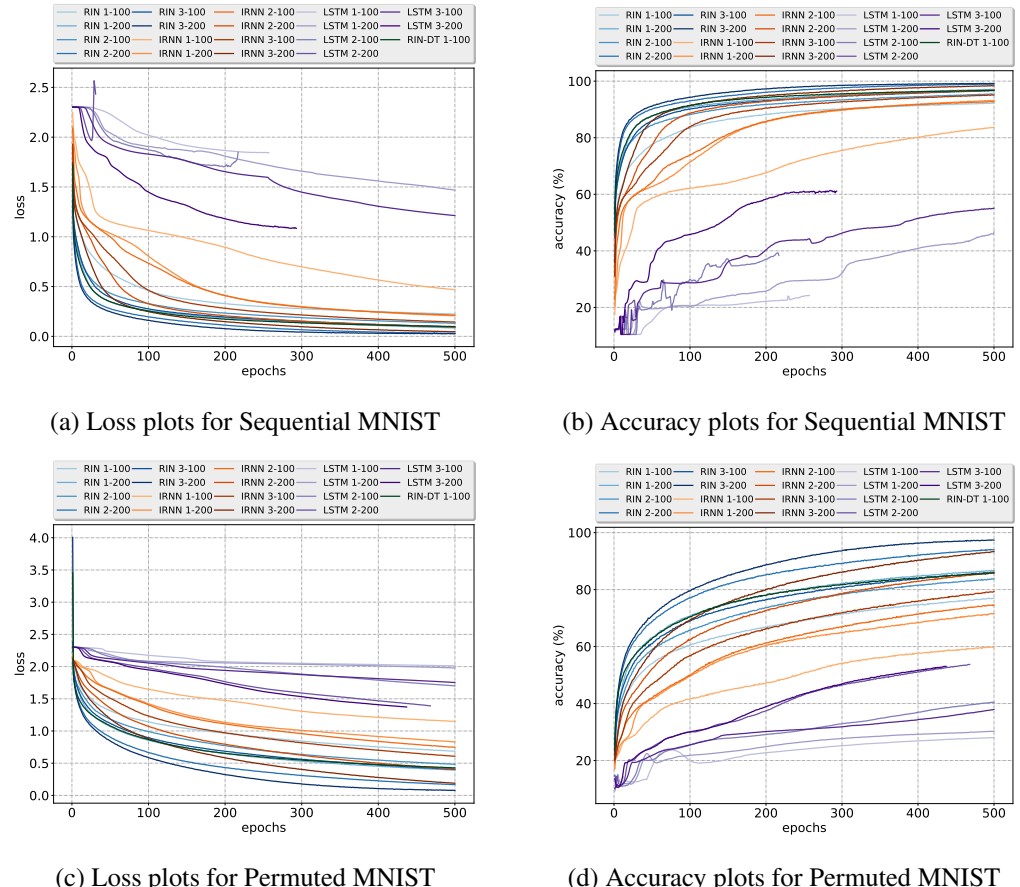

(a) Loss plots for Sequential MNIST

(b) Accuracy plots for Sequential MNIST

(c) Loss plots for Permuted MNIST

(d) Accuracy plots for Permuted MNIST

Figure 6: (a) and (b) show the training loss and accuracy plots for Sequential MNIST; (c) and (d) present training loss and accuracy plots for Permuted MNIST. We use blue color palette to represent RIN experiments, orange color palette to represent IRNN experiments, purple color palette to represent LSTM experiments and green color palette to represent RIN-DT experiments. RINs and RIN-DTs are much better than IRNNs and LSTMs in the early stage of training (first 200 epochs).

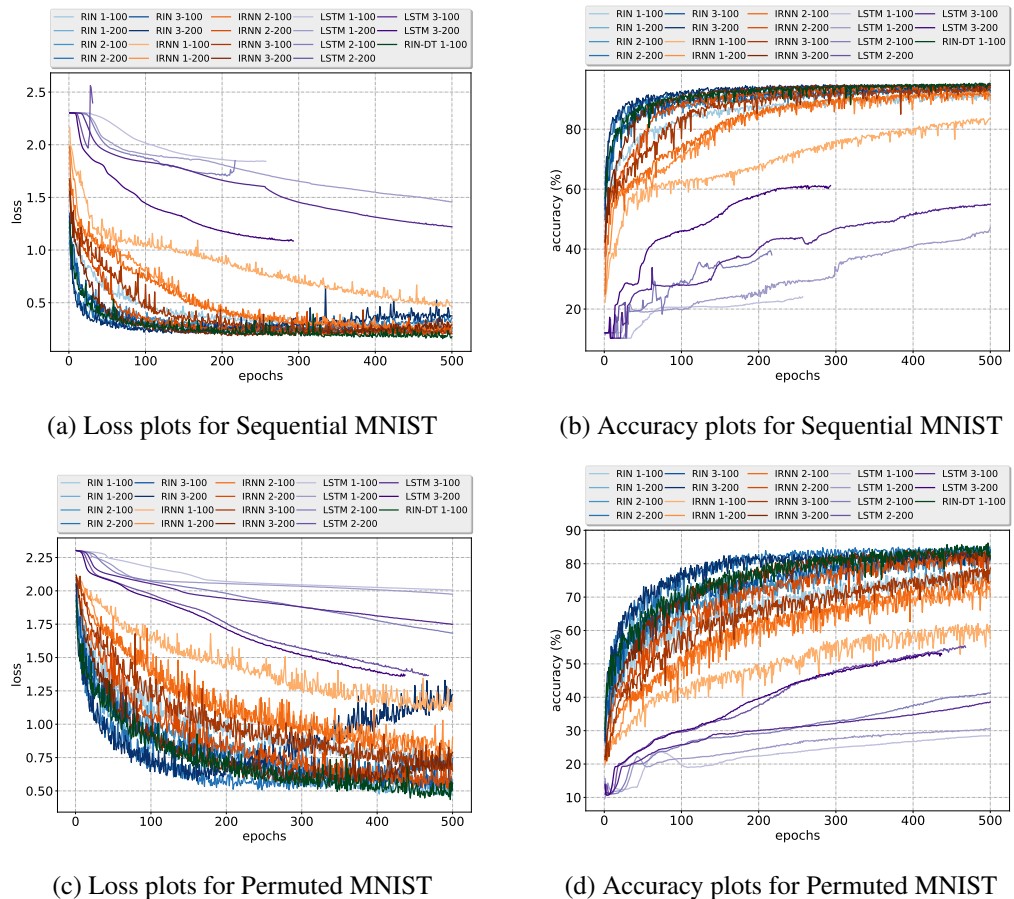

(a) Loss plots for Sequential MNIST

(b) Accuracy plots for Sequential MNIST

(c) Loss plots for Permuted MNIST

(d) Accuracy plots for Permuted MNIST

Figure 7: (a) and (b) show the test loss and accuracy plots for Sequential MNIST; (c) and (d) present test loss and accuracy plots for Permuted MNIST.

Fig. 8–12 show validation of the Iterative Estimation hypothesis in all network types.

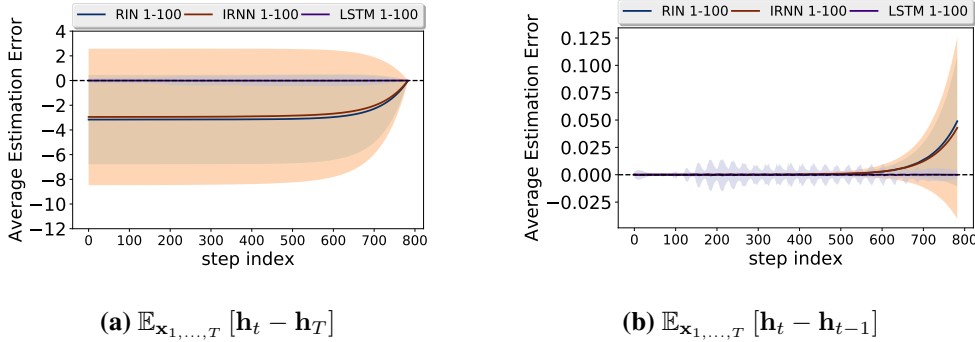

**(a)** $\mathbb{E}_{\mathbf{x}_{1,\dots,T}}\left[\mathbf{h}_t - \mathbf{h}_T\right]$        **(b)** $\mathbb{E}_{\mathbf{x}_{1,\dots,T}}\left[\mathbf{h}_t - \mathbf{h}_{t-1}\right]$

Figure 8: Evidence of learning identity mapping in RIN, IRNN and LSTM for network type 1–100 over a batch of 128 samples. **(a)** evaluates Eq. 1 and **(b)** evaluates Eq. 3.

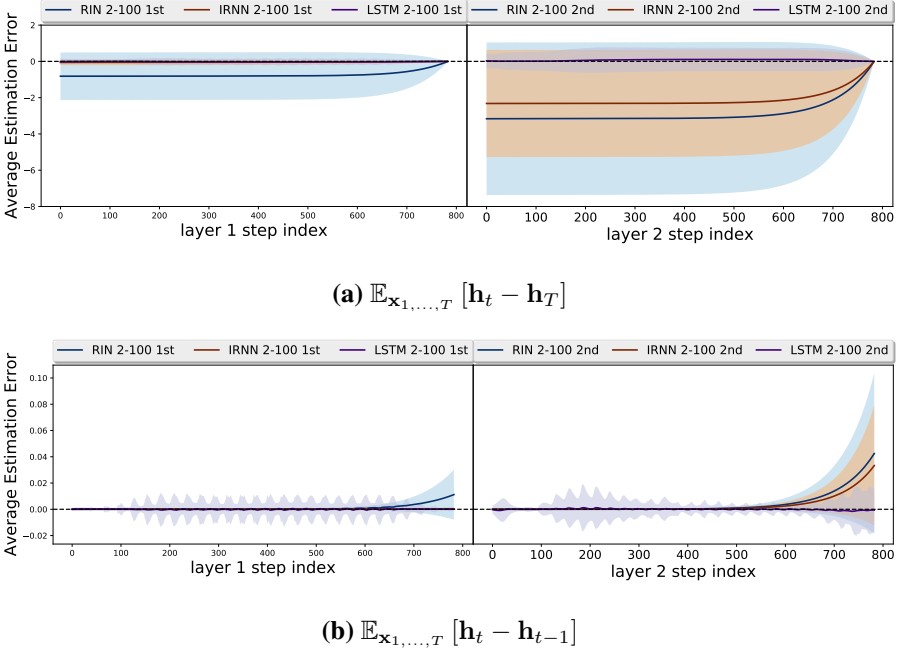

**(a)** $\mathbb{E}_{\mathbf{x}_{1,\dots,T}}\left[\mathbf{h}_t - \mathbf{h}_T\right]$

**(b)** $\mathbb{E}_{\mathbf{x}_{1,\dots,T}}\left[\mathbf{h}_t - \mathbf{h}_{t-1}\right]$

Figure 9: Evidence of learning identity mapping in RIN, IRNN and LSTM for network type 2–100 over a batch of 128 samples. **(a)** evaluates Eq. 1 and **(b)** evaluates Eq. 3.

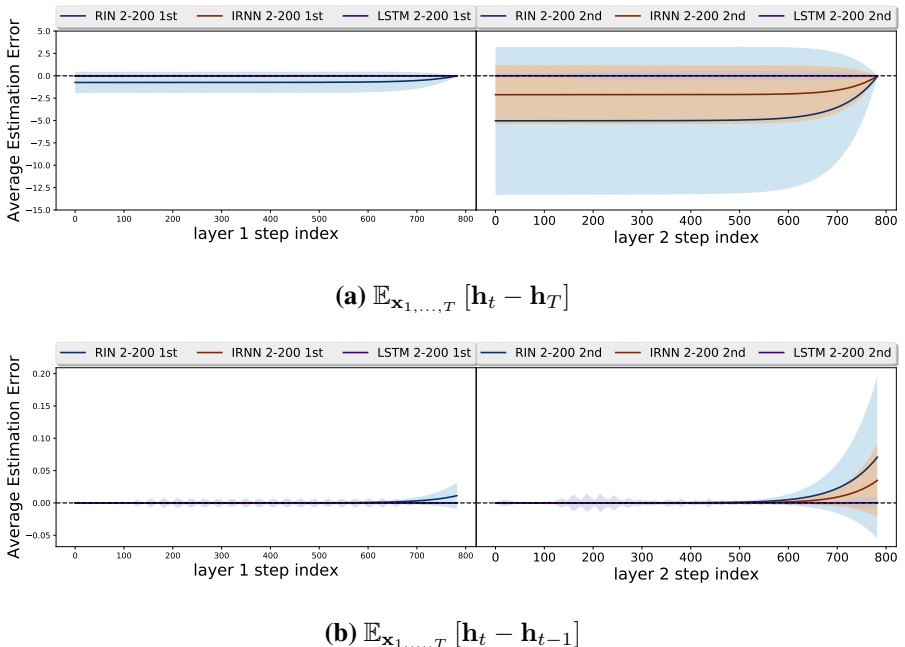

**(a)** $\mathbb{E}_{\mathbf{x}_{1,\ldots,T}}\left[\mathbf{h}_t - \mathbf{h}_T\right]$

**(b)** $\mathbb{E}_{\mathbf{x}_{1,\ldots,T}}\left[\mathbf{h}_t - \mathbf{h}_{t-1}\right]$

Figure 10: Evidence of learning identity mapping in RIN, IRNN and LSTM for network type 2–200 over a batch of 128 samples. **(a)** evaluates Eq. 1 and **(b)** evaluates Eq. 3.

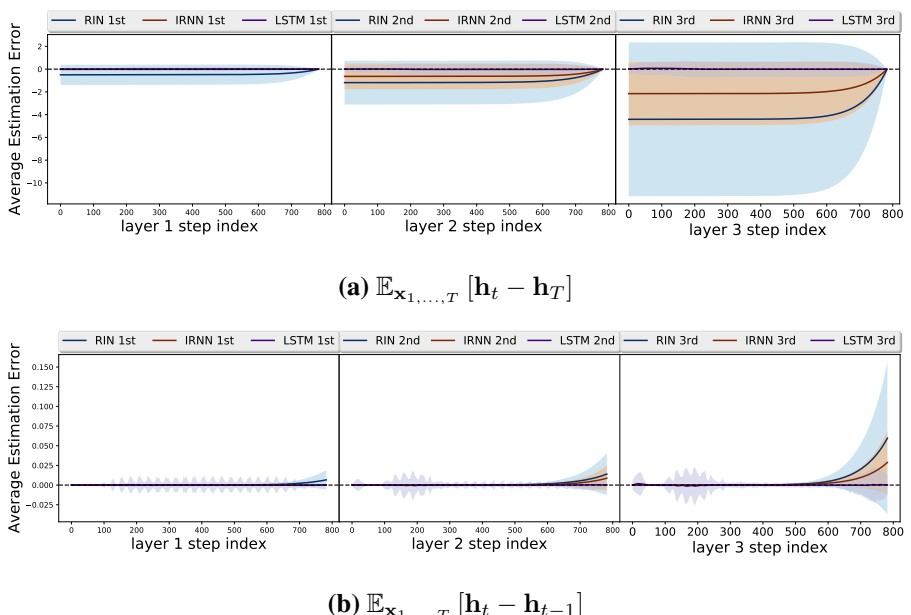

**(a)** $\mathbb{E}_{\mathbf{x}_{1,\ldots,T}}\left[\mathbf{h}_t - \mathbf{h}_T\right]$

**(b)** $\mathbb{E}_{\mathbf{x}_{1,\ldots,T}}\left[\mathbf{h}_t - \mathbf{h}_{t-1}\right]$

Figure 11: Evidence of learning identity mapping in RIN, IRNN and LSTM for network type 3–100 over a batch of 128 samples. **(a)** evaluates Eq. 1 and **(b)** evaluates Eq. 3.

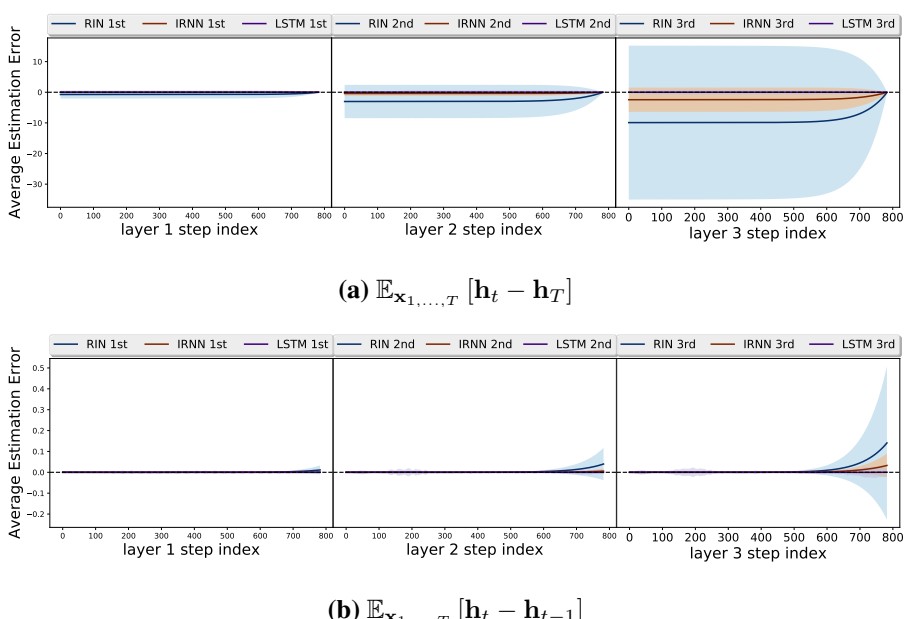

Figure 12: Evidence of learning identity mapping in RIN, IRNN and LSTM for network type 3–200 over a batch of 128 samples. **(a)** evaluates Eq. 1 and **(b)** evaluates Eq. 3.

# D    RINS WITH DEEP TRANSITIONS

In Section 3.2, we tested an additional model for RINs, which takes the concept of Deep Transition Networks (DTNs) Pascanu et al. (2013). Instead of stacking the recurrent layers, DTNs add multiple nonlinear transitions in a single recurrent step. This modification massively increases the depth of the network. In our RIN-DTs, the number of transition per recurrent step is two. Because the length of the sequence for Sequential and Permuted MNIST tasks is 784, RIN-DTs have the depth of $784 \times 2 = 1568$. The recurrent layer is defined in Eqs. 26–27.

$$\hat{\mathbf{h}}_t = f(\mathbf{W}_1\mathbf{x}_t + (\mathbf{U}_1 + \mathbf{I})\mathbf{h}_{t-1} + \mathbf{b}_1) \qquad (26)$$
$$\mathbf{h}_t = f((\mathbf{U}_2 + \mathbf{I})\hat{\mathbf{h}}_t + \mathbf{b}_2) \qquad (27)$$

