# OpenReview forum: "Overcoming the vanishing gradient problem in plain recurrent networks"
_ICLR.cc/2018/Conference — Reject_

### Official Review · AnonReviewer3 · 2017-11-26
**confusing analysis, little novelty**

**Rating:** 2
**Confidence:** 4

**Review:**

Here are my main critics of the papers:

1. Equation (1), (2), (3) are those expectations w.r.t. the data distribution (otherwise I can't think of any other stochasticity)? If so your phrase "is zero given a sequence of inputs X1, ...,T" is misleading.
2. Lack of motivation for IE or UIE. Where is your background material? I do not understand why we would like to assume (1), (2), (3). Why the same intuition of UIE can be applied to RNNs?
3. The paper proposed the new architecture RIN, but it is not much different than a simple RNN with identity initialization. Not much novelty.
4. The experimental results are not convincing. It's not compared against any previous published results. E.g. the addition tasks and sMNIST tasks are not as good as those reported in [1]. Also it only has been tested on very simple datasets.


[1] Path-Normalized Optimization of Recurrent Neural Networks with ReLU Activations. Behnam Neyshabur, Yuhuai Wu, Ruslan Salakhutdinov, Nathan Srebro.

---

> ### Author Response · Authors · 2017-12-07
> **Thanks for your insightful comments!**
>
> 1) Thanks for your comments. We have fixed this in the updated revision.
>
> 2) The observation in Fig. 1(a) and (b) for LSTM motivates us to study the iterative estimation view in RNNs. Initially, we thought that this phenomenon only exists in gated neural networks such as LSTM and GRU. After we analyzed the GNN (in section 2.2), we found that the gating mechanism is not the only way that an RNN can be trained deeply.
>
> 3) In the IRNN paper, the authors proposed to use the identity initialization the observation that “when the error derivatives for the hidden units are backpropagated through time they remain constant provided no extra error-derivatives are added”. In this paper, we view the proposal of RIN from a different direction where the “surrogate memory” component helps the network to learn identity mapping in which there is no help from the gates in RNNs. Additionally, we found in the experiments that the RIN is more stable and faster to train than IRNN.
>
> 4) Thank you for comments on this issue. We are aware that there are papers that produce higher scores in Sequential and Permuted MNIST. However, with the same experiment settings as in other paper, we cannot reproduce these numbers for the baseline models while developing this paper. We finally decided to report only our scores because the numbers reported in different works have a large variance. In this paper, we used a uniform experiment setting that compares the networks fairly and imposes as fewer constraints as possible. Thanks for pointing out this nice Path-SGD paper. Neyshabur et al. (2016) tackled the problem of using ReLU in RNNs from an optimization point of view. Similarly, Neyshabur et al. (2016) also found that IRNNs suffer from severe instability during training. This observation matches our results as well.

---

### Official Review · AnonReviewer1 · 2017-11-27
**Iterative Estimation point of view on recurrent GNN**

**Rating:** 4
**Confidence:** 5

**Review:**

The paper investigates the iterative estimation view on gated recurrent networks (GNN). Authors observe that the average estimation error between a given hidden state and the last hidden state  gradually decreases toward zeros. This suggest that GNN are bias toward an identity mapping and learn to preserve the activation through time.
Given this observation, authors then propose RIN, a new RNN parametrization where the hidden to hidden matrix is decomposed as a learnable weight matrix plus the identity matrix.
Authors evaluate their RIN on the adding, sequential MNIST and the baby tasks and show that their IRNN outperforms the IRNN and LSTM models.

Questions:
- Section 2 suggests that use of the gate  in GNNs encourages to learn an identity mapping. Does the average iteration error behaves differently in case of a tanh-RNN ?
- It seems from Figure 4 (a) that the average estimation error is higher for RIN than IRNN and LSTM and only decrease toward zero at the very end. What could explain this phenomenon?
- While the LSTM baseline matches the results of Le et al., later work such as Recurrent Batch Normalization or Unitary Evolution RNN have demonstrated much better performance with a vanilla LSTM on those tasks (outperforming both IRNN and RIN). What could explain this difference in the performances?
- Unless I am mistaken, Gated Orthogonal Recurrent Units: On Learning to Forget from Jing et al. also reports better performances for the LSTM (and GRU) baselines that outperform RIN on the baby tasks with mean performances of 58.2 and 56.0 for GRU and LSTM respectively?

- Quality/Clarity:
The paper is well written and pleasant to read

- Originality:
Looking at RNN from an iterative refinement point of view seems novel.

- Significance:
While looking at RNN from an iterative estimation is interesting, the experimental part does not really show what are the advantages of the propose RIN. In particular, the LSTM baseline seems to weak compared to other works.

---

> ### Author Response · Authors · 2017-12-07
> **Thank you very much for your thoughtful comments!**
>
> 1) We assume the tanh-RNN is a vanilla RNN with tanh activation. We run an experiment on tanh-RNN with the adding problem. The network failed to converge. Both average estimation error and variance stay very close to zero (<10^-5). The difference between steps, in this case, is not informative and nearly noise.
>
> 2) Thanks for your comments. Indeed the RIN’s average estimation error in Fig. 4(a) is higher than the other two architectures. We believe that this is due to the choice of ReLU activation in RIN and IRNN. Repeated application of ReLU could cause this problem (Jastrzebski et al., 2017). However, the experiment results don’t suggest that larger average estimation errors lower the performance of RIN.
>
> 3) Thanks for pointing this out. In this paper, we do not claim that RIN is superior to LSTM. We tried to compare all three networks fairly with a uniform hyperparameter setting. We are aware that other papers produce higher scores in sequential and permuted MNIST. However, even with the same experiment settings (at best of our knowledge), we are unable to reproduce these numbers for sequential and permuted MNIST during the development of this paper. We finally decided to report only our numbers because there is a large variance of reported scores by different previous works.
>
> 4) Thank you very much for pointing this out. Sadly, we only found out that the authors updated the numbers for bAbI tasks after the submission of our paper (the 3rd version is updated on Oct 25th). The numbers in the paper are taken from the 2nd version of the paper, and we did our best to replicate their experiment settings (regarding the network architecture). Note that the description for the bAbI tasks is very brief, they did not reveal the training procedure even in the 3rd version.
>
> S. Jastrzebski, D. Arpit, N. Ballas, V. Verma, T. Che, and Y. Bengio. Residual Connections Encourage Iterative Inference. CoRR, abs/1710.04773, October 2017.

---

> > ### Comment · AnonReviewer1 · 2018-01-12
> > **Comparison with previous works**
> >
> >  Thanks for your reply and clarifications.
> >
> > I think overall this is a very interesting direction.
> > However,   authors did not address the comparison with previous work in their paper (weak LSTM baselines). This is of importance to properly evaluate  the main contribution of this work. Therefore, I have decided to revise my rating slightly down.

---

### Official Review · AnonReviewer2 · 2017-11-29
**Simple trick to improve the training of simple RNNs. Novel idea and well presented, but the experimental evaluation could be improved**

**Rating:** 7
**Confidence:** 4

**Review:**

Summary:
The authors present a simple variation of vanilla recurrent neural networks, which use ReLU hiddens and a fixed identity matrix that is added to the hidden-to-hidden weight matrix. This identity connection acts as a “surrogate memory” component, preserving hidden activations over time steps.
The experiments demonstrate that this architecture reliably solves the addition task for up to 400 input frames. It also achieves a very good performance on sequential and permuted MNIST and achieves SOTA performance on bAbI.
The authors observe that the proposed recurrent identity network (RIN) is relatively robust to hyperparameter choices. After Le et al. (2015), the paper presents another convincing case for the application of ReLUs in RNNs.

Review:
I very much like the paper. The motivation and architecture is presented very clearly and I am happy to also see explorations of simpler recurrent architectures in parallel to research of gated architectures!
I have a few comments and questions:
1) Clarification: In Section 2.2, do you really mean bit-wise multiplication or element-wise? If bit-wise, can you elaborate why? I might have missed something.
2) Why does the learning curve of the IRNN stop around epoch 270 in Figure 2c? Also some curves in the appendix stop abruptly without visible explosions. Were these experiments run until completion? If so, would it be possible to plot the complete curves?
3) I think for a fair comparison with LSTMs and IRNNs a limited hyperparameter search should be performed separately on all three architectures at least for the addition task. Optimal hyperparameters are usually model-specific. Admittedly, the authors mention that they do not intend to make claims about superior performance to LSTMs, however the competitive performance of small RINs is mentioned a couple of times in the manuscript.
Le et al. (2015) for instance perform a coarse grid search for each model.
4) I wouldn't say that ResNets are Gated Neural Networks, as the branches are just summed up. There is no (multiplicative) gating as in Highway Networks.
5) I think what enables the training of very deep networks or LSTMs on long sequences is the presence of a (close-to-)identity component in forward/backward propagation, not the gating. The use of ReLU activations in IRNNs (with identity initialization of the hidden-to-hidden weights) and RINs (effectively initialized with identity plus some noise) makes the recurrence more linear than with squashing activation functions.
6) Regarding the absence of gating in RINs: What is your intuition on how the model would perform in tasks for which conditional forgetting is useful. Consider for example a task with long sequences, outputs at every time step and hidden activations not necessarily being encouraged to estimate last step hidden activations. Would RINs readily learn to reset parts of the hidden state?
7) Henaff et al. (2016) might be related, as they are also looking into the addition task with long sequences.

Overall, the presented idea is novel to the best of my knowledge and the manuscript is well-written. I would recommend it for acceptance, but would like to see the above points addressed (especially 1-3 and some comments on 4-6). After a revision I would consider to increase the score.

References:
Henaff, Mikael, Arthur Szlam, and Yann LeCun. "Recurrent orthogonal networks and long-memory tasks." In International Conference on Machine Learning, pp. 2034-2042. 2016.
Le, Quoc V., Navdeep Jaitly, and Geoffrey E. Hinton. "A simple way to initialize recurrent networks of rectified linear units." arXiv preprint arXiv:1504.00941 (2015).

---

> ### Author Response · Authors · 2017-12-07
> **Thanks for your nice review! We address your questions as follows**
>
> 1) Thanks for pointing this out. It should have been element-wise instead of bit-wise. We’ve fixed this in the updated revision.
>
> 2) We employed Early Stopping during the training. The reason for the unfinished LSTM experiments is because the overfitting occurred. The unfinished IRNN experiment is because the training is interrupted by the explosion of the training error (see Fig. 5(c) for training curve). We tried to mitigate this problem by imposing a relative loose gradient clipping (100), in the end, IRNN is still unstable if the sequence is long (for example in T3).
>
> 3) Thanks for the comments. We try to select a set of hyperparameters that can offer a fair comparison of all three tested networks. In preliminary experiments, we have tried different learning rates from {10^−2, 10^−3, 10^−4, 10^−5, 10^−6}. We chose the largest learning rate that does not cause training failure for IRNNs.
>
> 4) Thanks for the comments on residual networks. It is true that ResNets do not have multiplicative gates as in Highway Networks. In this paper, we view ResNets as a subcase of Highway Networks where the gates are fully open as pointed out by Greff et al. 2016.
>
> 5) Thanks for your comments. We felt the same way for training with long sequences. However, the gating mechanism may be very important in tasks that desire to regulate the network and provide explicit control for hidden activations.
>
> 6) There is no mechanism for the network to perform explicit conditional forgetting. RIN may not be capable of readily resetting its hidden states. We will perform more experiments to determine when the network would fail on tasks with long-sequence.
>
> 7) Thanks for pointing this interesting article.
>
> Klaus Greff, Rupesh Kumar Srivastava, and Jürgen Schmidhuber. Highway and residual networks learn unrolled iterative estimation. CoRR, abs/1612.07771, 2016.

---

> > ### Comment · AnonReviewer2 · 2018-01-12
> > **response to rebuttal**
> >
> > After reviewing the revised draft, I have decided to not increase the score. I think 7 is still appropriate, as I'm not too sure about the impact.

---

### Decision · Program_Chairs · 2018-01-29
**ICLR 2018 Conference Acceptance Decision**

**Decision:**

Reject

**Comment:**

The authors propose to use identity + some weights in the recurrent connections to prevent vanishing gradients. The reviewers found the experiments to have weak baselines, weakening the claims of the paper.